# Zebrafish as an Emerging Model for Sarcopenia: Considerations, Current Insights, and Future Directions

**DOI:** 10.3390/ijms242317018

**Published:** 2023-11-30

**Authors:** Santiago Callegari, Foad Mirzaei, Lila Agbaria, Sanobar Shariff, Burhan Kantawala, Desmond Moronge, Brian M. O. Ogendi

**Affiliations:** 1Vascular Medicine Outcomes Laboratory, Cardiology Department, Yale University, New Haven, CT 06510, USA; 2Faculty of General Medicine, Yerevan State Medical University after Mikhtar Heratsi, 2 Koryun, Yerevan 0025, Armenia; foadmirzaee59@gmail.com (F.M.); lalamu165@gmail.com (L.A.); burhan.kantawala@gmail.com (B.K.); 3Department of Physiology, Medical College of Georgia, Augusta, GA 30912, USA; dmorongue@gmail.com; 4Department of Medicine, Michigan State University College of Human Medicine, Grand Rapids, MI 49503, USA; bogendi@gmail.com

**Keywords:** sarcopenia, zebrafish, age-related muscle deficits, aging, biomarkers, electrical impedance myography

## Abstract

Sarcopenia poses a significant challenge to public health and can severely impact the quality of life of aging populations. Despite extensive efforts to study muscle degeneration using traditional animal models, there is still a lack of effective diagnostic tools, precise biomarkers, and treatments for sarcopenia. Zebrafish models have emerged as powerful tools in biomedical research, providing unique insights into age-related muscle disorders like sarcopenia. The advantages of using zebrafish models include their rapid growth outside of the embryo, optical transparency during early developmental stages, high reproductive potential, ease of husbandry, compact size, and genetic tractability. By deepening our understanding of the molecular processes underlying sarcopenia, we may develop novel diagnostic tools and effective treatments that can improve the lives of aging individuals affected by this condition. This review aims to explore the unique advantages of zebrafish as a model for sarcopenia research, highlight recent breakthroughs, outline potential avenues for future investigations, and emphasize the distinctive contributions that zebrafish models offer. Our research endeavors to contribute significantly to address the urgent need for practical solutions to reduce the impact of sarcopenia on aging populations, ultimately striving to enhance the quality of life for individuals affected by this condition.

## 1. Introduction

Zebrafish (*Danio rerio*) has emerged as a highly promising animal model in the realm of biomedical research, offering invaluable insights into a diverse array of human diseases, including but not limited to skeletal muscle atrophy, aging, and sarcopenia. This diminutive aquatic organism has earned its place as an indispensable model organism for investigating crucial pathways and therapeutics associated with various facets of human health, such as neurodegenerative disorders, skeletomuscular ailments, and cardiovascular conditions [1,2,3,4]. Genetically, the zebrafish genome shares an astonishing 70% similarity with the human genome, with approximately 82% of genes associated with human ailments having identifiable orthologues in zebrafish [5,6].

Primary sarcopenia, a condition characterized by the age-related loss of muscle mass and function, which significantly compromises the quality of life in affected individuals, has been extensively studied utilizing model organisms, including mice, rats, flies, and worms. These models have facilitated the identification of therapeutic strategies and genetic modifications aimed at ameliorating muscle-related degenerative processes in aging individuals [6]. However, practical diagnostic modalities, accurate biomarkers for risk stratification or follow-up, and effective treatments for primary sarcopenia are still elusive. Zebrafish models offer a variety of muscle growth mechanisms, bridging the evolutionary gap between fly and mouse models. Despite the apparent senescence and fixed lifetime of mammalian skeletal muscle, fish tissues maintain significant plasticity even at the adult stage. This allows researchers to examine the metabolic functions such as peroxisome-resident proteins using morpholino-mediated knockdown or transcription activator-like effector nucleases [6,7,8].

Zebrafish, in particular, has assumed a significant role in biochemical, physiological, genetic, and developmental research due to several inherent advantages: (i) rapid ex utero growth, with adulthood attained within a mere three months, (ii) the optical transparency of embryos and early larvae, (iii) high reproductive potential, (iv) compact size of approximately 1.5 inches when fully matured and a short generation time, (v) facile husbandry and reproduction, and (vi) the ability for genetic manipulation [9,10,11,12] (Figure 1).

The primary objective of this study is to elucidate the manifold advantages of employing zebrafish as a model for sarcopenia, shedding light on its resemblances and disparities relative to the human skeletal muscle. This manuscript will also critically evaluate recent breakthroughs in the field, delineate potential avenues for future research, and underscore the distinctive contributions that the zebrafish model offers in comparison to other fish models.

This literature review utilizes a broad-scope methodology involving systematic database searches on PubMed, Scopus, and Web of Science. The search involves looking for specific terms related to sarcopenia and zebrafish as well as age-related muscle deficits in the title and abstract of an article. Additionally, the databases were explored using MeSH terms from the three categories mentioned above. The identified research articles were then assessed for relevance and quality.

## 2. Zebrafish as an Animal Model of Primary Sarcopenia

Within the context of skeletal muscle research, zebrafish exhibits striking parallels with the human skeletal muscle system, including the presence of satellite-like cells responsible for muscle repair and regeneration, and the anatomical distinction between fast and slow muscles. The fast muscles of zebrafish are situated around the vertebrae, while the slow muscles reside beneath the body’s surface [9,10,11,12]. Moreover, zebrafish harbors a collection of orthologous genes analogous to human Myogenic Regulatory Factors (MRFs) [4]. Skeletal muscles, constituting a substantial portion (35–50%) of the human body’s volume, play an integral role in physical mobility and critical metabolic processes, such as basal metabolic rate, glucose uptake, and lipid metabolism [3,4]. The maintenance of muscle mass within myofibers hinges on a delicate balance between protein synthesis and breakdown, with disruptions in this equilibrium resulting in reduced muscle mass and myofiber size, and subsequently, metabolic disorders and muscular atrophy [13].

Each animal model in sarcopenia research has its own set of advantages and disadvantages. Rodents, which mimic human muscle, allow for effective therapy trials and regeneration inquiry; however, they have cost and sample size limitations. Drosophila and Caenorhabditis elegans, while inexpensive and informative in genetics, lack human muscle similarity and cannot be utilized in co-morbidity investigations [9]. Despite the limited earlier research and co-morbidity study restrictions, zebrafish combines cost-effectiveness and longevity, offering an opportunity to make new discoveries, especially using the available high-output platforms.

Numerous investigations have been carried out to explore the signaling networks involved in muscle metabolism in zebrafish and their resemblances to humans. Walters and colleagues, for instance, concentrated on the relationship between muscle degeneration, leukocyte infiltration, and apoptosis [14]. Furthermore, another study investigated the intricate regulatory mechanisms governing protein turnover during skeletal muscle development and disease, focusing on Kelch Like Family Member 40 (KLHL40), an E3 ubiquitin ligase cullin3 (CUL3) substrate-specific adapter protein. The zebrafish study reveals that KLHL40, a key regulator of skeletal muscle development, exerts influence over ER-Golgi anterograde trafficking and Sar1a degradation. Deficiency in KLHL40 results in structural and functional abnormalities, disrupting ER exit site vesicle formation and the transport of extracellular cargo proteins, providing insightful perspectives into disease mechanisms [15].

Additionally, the investigation into cryotherapy uncovers the molecular effects of acute cold exposure on skeletal muscle, unveiling a significant post-cryotherapy upregulation of genes associated with “protein ubiquitination”. Temporal analysis indicates a transient enhancement in the expression of E3 ubiquitin ligases, notably the muscle RING-finger protein-1 (MuRF1). This temporal modulation offers valuable information about the potential advantages and risks associated with cryotherapy for the management of muscle trauma or fatigue, revealing a temporary increase in the expression of E3 ubiquitin ligases, particularly MuRF1 [16]. These results of zebrafish research provide a template for personalized treatment in muscle-related disorders such as sarcopenia. Targeted therapeutics can be developed by using zebrafish models to uncover major genetic regulators and molecular mechanisms driving muscle growth and illness. These findings pave the way for the development of precision treatments that target specific genetic variants, possibly altering how we approach customized care for individuals with sarcopenia.

In light of the remarkable progress achieved in recent years, zebrafish has bolstered its status as a potential model for elucidating the molecular underpinnings of sarcopenia and conducting rigorous therapeutic screening. Recent research exemplifies the incorporation of cutting-edge methodologies, such as electrical impedance myography, showcasing the substantial advancements within this field.

## 3. Current Challenges and Opportunities

### 3.1. Aging and Sarcopenia

As we age, our bodies endure a variety of changes. If any of these alterations continue for an extended period, they may contribute to the development of various illnesses [17]. Despite the possible participation of various molecular pathways in the aging process, we still do not fully comprehend its complexities [18]. Many nutritional (such as vitamins and minerals) and hormonal deficits have been identified during natural aging, and these shortcomings are suspected to be contributing factors to chronic diseases that are particularly prevalent among the elderly (such as sarcopenia and neurodegenerative disease) [19,20]. Among the nutritional factors are magnesium, zinc, vitamin D, vitamin B, etc.; however, the specific link between these deficits and illnesses in elderly individuals is doubtful [19,21,22,23,24]. In the context of aging, several investigations in various zebrafish tissues (such as cardiac, neural, and skeletomuscular tissues) have been implemented to find the fundamental mechanisms of aging due to its advantages over other animal models and strong similarities to humans [3,25,26,27].

Sarcopenia is defined as the chronic loss of skeletal muscle mass and efficiency throughout the body and can be categorized into two forms: primary (age-associated) and secondary (illness-associated) [28,29]. Primary sarcopenia is caused by aging alone, whereas secondary sarcopenia occurs in the presence of other disorders along with aging [30,31]. This age-related muscle atrophy may cause falls and subsequent fractures, osteoporosis, frailty, metabolic shifts, increased heart disease, immunodeficiency, and various other disorders among elderly individuals [32,33].

Aging results in alterations in muscle mass and function in virtually everyone, but the rapidity at which these changes occur varies from person to person with a multitude of factors, the majority of which are still unknown (Figure 2) [34,35] (Figure 1). Its prevalence in the elderly population can vary from 10% to 27% [36], and with the increasing elderly population, it is predicted to become a significant health issue by 2030 [37]. Revealing the economic facet, approximately USD 40.4 billion is the potential hospitalization cost of sarcopenia patients [38]. In this context, zebrafish presents a distinct path of hope; their potential utility in unraveling the complexity of sarcopenia and its underlying processes offers a viable path for cutting-edge treatments and therapeutic approaches.

### 3.2. Pathogenesis of Sarcopenia

Sarcopenia has been associated with several processes, including oxidative stress, mitochondrial dysfunction, senescent satellite cell accumulation, protein homeostasis imbalance, inappropriate diet, NAD+ deterioration, increased myostatin and GDF-15, immunosenescence, gut microbiota alteration, and hormonal deviations [29,30,39,40,41,42,43] (Figure 3). Despite extensive investigations in these areas, more detailed research efforts are still needed. As zebrafish gains increasing momentum in the field of aging, several different zebrafish models and experiments will be performed in these areas to validate and further expand the animal model with different strains and high-output platforms. Other possible biomarkers for cancer, disorders of neurological development, hepatotoxicity, and other conditions also have been identified in zebrafish, providing novel insights into the identification of molecular pathways and the early detection of these pathologies [44,45,46].

Sustained oxidative stress may contribute to the progression of sarcopenia by influencing synaptic junctions (reducing acetylcholine and synaptic effectiveness), protein balance (increased protein degradation and decreased protein synthesis), and most importantly, mitochondria, which are the main sources of ROS [47,48]. Dysfunctional mitochondria are proposed to be caused by the inactivation of the SIRT1-PGC1α axis, crucial for mitochondrial biogenesis [49]. Subsequently, the accumulation of persistent oxidative damage and the resulting mitochondrial malfunction coincide with the potential of telomere breakage, creating an environment permissive of cellular senescence [41].

Satellite cells (SCs) in skeletal muscle have regenerative capabilities, but this ability may diminish with age due to various factors (such as telomere shrinkage, defective mitochondria, so on), leading to cellular senescence [50,51]. Different regulatory pathways, including p53-p21CIP1-CDK2 and p16Ink4a-CDK4/6, play a role in the initiation of senescence [52,53]. The contribution of p53 to sarcopenia is still debated [54]. The total count of SCs during aging remains uncertain due to conflicting findings [55]. As previously noted, hyperplastic development persists in zebrafish skeletal muscle throughout life (unlike humans) due to specific cells that generate Pax7, similar to human satellite cells. These Pax7+ cells were additionally identified to be situated similarly to human SCs between the basement membrane and the sarcolemma [56]. Further investigations uncovered a zebrafish ortholog to mammalian FOXM1 that may have comparable functions in muscle tissue maintenance such as cellular senescence regulation [57]. These fish also can be employed to investigate the association between telomere shrinkage and sarcopenia as zebrafish and human telomeres are remarkably comparable [58]. Zebrafish models have been developed to simulate age-related disorders, including telomere malfunction, lipofuscin buildup, vitamin E depletion, and obesity-associated sarcopenia [59,60,61].

Protein homeostasis (production and breakdown) in skeletal muscle includes numerous mechanisms, and its disruption contributes to the occurrence of sarcopenia [62]. Myostatin, synthesized by various tissues, including skeletal muscle, suppresses protein synthesis by inhibiting the AKT-mTOR axis after binding to the activin type IIB receptor [63]. Increased myostatin has been associated with aging and is considered a possible mechanism for sarcopenia development [64]. The deactivation of the AKT-mTOR axis not only suppresses protein synthesis but also accelerates protein degradation through the elimination of AKT’s inhibitory impact on Forkhead box O (FoxO) [63]. Notably, myostatin depletion in mutant zebrafish resulted in a growth in muscle tissue as well as a metabolic shift toward lipid utilization. Although the particular mechanism of myostatin on skeletal muscle differs across humans and zebrafish in some domains, the study provides insight into the general molecular interplay throughout metabolic alterations and muscle development [65].

In conclusion, several processes, such as long-term low-level inflammation, hormonal shifts, cellular senescence, mitochondrial failure, altered myokines, and others, may contribute to sarcopenia, but its precise root cause remains unknown. Although a substantial number of experiments are required to unravel these intricate phenomena, using zebrafish as a high-output potent model has the capacity to elucidate the intricate in vivo mechanisms underlying the development of sarcopenia, facilitating the identification of novel pathways and mediators essential for an in-depth understanding of these intricate mechanisms.

### 3.3. Diagnosis of Sarcopenia

Sarcopenia is diagnosed by assessing muscle quantity (mass) and quality (strength and performance). Various tests are used for strength analysis, physical performance evaluation, and assessing muscle mass through imaging technologies [28,66]. Despite the available assessments, there are limitations in specific diagnostic domains. Certain exercise or nutrition programs may show a beneficial impact on some evaluations but not others, reducing their reliability [67,68]. Thus, improving diagnostic tools and procedures is urgent.

The decline in skeletal muscle size and strength is not proportional. Studies show that muscle mass decreases by 6% every ten years after mid-age, while strength declines faster, and the rate is twice as high in men compared to women [69,70]. These findings indicate measuring mass alone may not reliably detect sarcopenia in its early stages. Previous investigations suggested that reduced muscle mass accounts for a small percentage of the decline in muscular strength, and other areas require additional research since, in particular cases, conserving or improving muscle mass did not result in the anticipated rise in muscle strength [69].

Henderson et al. used the sarcopenic zebrafish animal model to address the complex interplay between muscle degeneration due to aging, and oxidative damage resulting from vitamin E deficiency, particularly in the context of skeletal muscle metabolism. The study employs metabolomics to analyze the skeletal muscle of aging zebrafish subjected to a long-term vitamin E deficiency. It reveals distinct metabolic changes associated with aging and vitamin E deficiency, indicating overlapping but unique alterations in metabolic pathways and underscoring the need for further comprehensive investigation using confirmatory approaches to understand their combined impact on muscle health [59]. Several biomarkers have been proposed to diagnose sarcopenia early, including inflammation-related biomarkers (TNF-α and CRP), nutritional-related biomarkers (insulin-like growth factor-1 and vitamin D), and genetic biomarkers (Rho GTPase activating protein 36 and family sequence similarity 171, member protein A1) [71,72,73]. However, there is no gold standard biomarker for its etiology, requiring further research. Carbonic anhydrase 2 (ca2), selenium-binding protein 1 (selenbp1), and myosin regulatory light chain 2 a and b (myl2a/b) have been proposed as age markers in zebrafish due to their striking resemblance in function and structure in their mammalian orthologs. It should be emphasized that age biomarkers may be altered in opposite ways in various tissues, for example, cardiac myl2b is elevated till the middle age in contrast to muscle myl2b in zebrafish [74].

The diagnosis of sarcopenia continues to be challenging due to the lack of standardized practical criteria, extensively validated biomarkers, or simple diagnostic modalities available for primary care. As we progressively improve our understanding of age-related muscle deficits, the discovery of novel mechanistic pathways and the validation of potential technologies (such as electrical impedance myography) could be accomplished with the recently introduced zebrafish animal models.

### 3.4. Treatment

Resistance training (RT) with a protein-rich diet is considered an effective starting point for managing age-associated skeletal muscle degradation [75,76]. Different exercise variables and the patient’s general health condition complicate implementing research findings in clinical practice [77,78], requiring researchers to develop ideal exercise instructions for sarcopenia therapy [79]. Additionally, exercise training in zebrafish has been found to lead to white muscle fiber hypertrophy and increased expression of genes involved in muscle synthesis and energy metabolism [80]. Aged zebrafish exhibit deficits in swimming performance due to weaker and shorter movements, indicating impairments in motor function [81].

Nutritional supplementation also plays a crucial role in sarcopenia management [75]. For example, vitamin D supplementation has shown conflicting outcomes in different studies, necessitating further investigation [22]. Whey protein peptide intake has been advantageous in both the prevention and management of sarcopenia [82,83]. Similarly, essential amino acid supplementation might trigger myocyte renewal and specialization [84,85].

Combining exercise with a healthy diet can induce mitochondrial biogenesis, increase satellite cell activity, and inhibit inflammatory cytokines, resulting in enhanced protein synthesis and decreased protein breakdown [86].

Pharmacological therapies are also considered for patients unable to undergo exercise interventions [87]. Although several therapies have been suggested, there are currently no FDA-approved medications for sarcopenia [88]. In an interesting study, the protective properties of Rhus coriaria extracts on cells subjected to oxidative damage were explored. It was shown that both human myoblast and zebrafish embryos demonstrated a slowing or halting of muscle atrophy [89]. This highlights the significance of using these fish in the development of pharmaceuticals.

In addition to the techniques described above, stem cell and mitochondrial transplantation have demonstrated an improvement in muscle mass and strength [55,90]. However, these approaches face challenges, such as the difficulties in isolation and transferring procedures, which may result in transferring malfunctioning mitochondria and be detrimental to the tissue. Moreover, the specific molecular effects of the transferring paths are not well known [91]. Therefore, further exploration and improvement are needed to successfully treat this disorder using these approaches.

In summary, given the existing lack of sufficient sarcopenia management and prevention measures, the need to discover and deliver effective approaches to address this issue remains unabated. As zebrafish offers a cost-effective, simple, high-output platform for the study of sarcopenia, the potential of chemical screening for discovering novel treatments using this model is significant. In addition, beyond the molecular pathways, the study of the mechanical function of the muscle and the gradual decline in muscle function in zebrafish similar to humans [81] convey significant potential for evaluating molecules in vivo to assess their efficacy and safety.

## 4. Latest Zebrafish Advances in Sarcopenia

Recent evidence suggests electrical impedance myography (EIM) as a feasible screening tool for skeletal muscle age-associated illnesses due to its affordability, simplicity of use, and ease of transport over other imaging tools [92]. EIM relies on the composition and subsequent the conductivity and electrical activity of the targeted muscle since it has been demonstrated that muscle conductivity and derived charge can alter in the presence of additional variables such as muscular atrophy and edema [93]. EIM has been performed on zebrafish using two separate techniques (surface and needle) to examine muscle tissue characteristics. Although the core principles of both approaches are the same, they differ in terms of accuracy and invasiveness. In general, when compared to alternative methods such as histological analysis and physical performance assessments in the study of zebrafish muscle tissue, EIM has proven to be less invasive, providing faster results and a greater ease of application and establishing its significance for sarcopenia research [81,94].

Rutkove et al. emphasized the lack of pharmacological treatments for sarcopenia and the impracticality of high-throughput chemical screens in mammalian models (Table 1). A novel surface electrical impedance myography (sEIM) platform has been lately introduced to quantitatively and noninvasively assess skeletal muscle health in adult zebrafish, effectively detecting muscle atrophy in aged zebrafish and correlating these measurements with established morphometric parameters. This demonstrates the utility of sEIM in studying the genetic factors contributing to muscle atrophy, offering potential applications as a noninvasive “virtual biopsy” in zebrafish for muscle and gerontology research. The study also investigated the impact of genetic deletion of gpr27, an orphan G-protein coupled receptor (GPCR), on skeletal muscle atrophy in aged zebrafish. Surface electrical impedance myography (sEIM) and histological analysis revealed that the absence of gpr27 exacerbated muscle atrophy, highlighting its role in modulating muscle health in aging animals [94].

Exercise has been the cornerstone of treatment to prevent the onset/progression and treat sarcopenia. This has also been evaluated in zebrafish. Exercise is known to effectively delay or prevent the onset and progression of sarcopenia, yet its underlying molecular mechanisms remain poorly understood. Sun et al. have utilized aged zebrafish as a valuable model to elucidate the pathological changes associated with sarcopenia, including reduced muscle fiber cross-sectional area, imbalanced protein synthesis and degradation, increased oxidative stress, and mitochondrial dysfunction. After subjecting these zebrafish to eight weeks of exercise intervention, the attenuation of these pathological changes was observed, particularly in relation to the activation of the AMPK/SIRT1/PGC-1α axis and the downregulation of 15-PGDH, providing potential therapeutic targets for the treatment of age-related sarcopenia through exercise intervention. Thus, with the ZF model, the benefit of exercise in terms of mitochondrial function was confirmed [95].

## 5. Future Perspectives

### 5.1. Novel Pathways and Molecules for Improving Mitochondrial Function

Sarcopenia and aging have been associated with progressive mitochondrial dysfunction that is characterized by deficiencies in mitochondrial content within the muscle in addition to decreased mitochondrial biogenesis, impaired mitochondrial permeability transition pore function, and increased mtDNA mutations [95,97]. Zebrafish is an excellent model to study muscle-related degenerations and sarcopenia due to their short lifespan and gradual aging like humans, while having conserved aging markers as in humans [98]. For instance, the zebrafish myosepta mimic the human tendons, both of which decline with age; therefore, the zebrafish model can be a perfect tool for studying mitochondrial tendon biomechanics and aging [98,99]. Due to these similarities, exercise intervention regimens related to mitochondria can be applied in the zebrafish. For example, it has been demonstrated that exercise significantly improved the expression of mitochondrial function biogenesis-related proteins while decreasing the expression of mitochondrial fission-related proteins in the zebrafish model [95]. In another study, aerobic exercise-induced in zebrafish, which repressed miR-128, has been reported to impair mitochondrial biogenesis by inhibiting PGC-1α [61,96]. Another study corroborated the importance of zebrafish as a model in identifying potential novel pathways for mitochondrial function. Here, the findings revealed that a high-fat diet reduced exercise capacity in a zebrafish model of obesity through pathways of decreased mitochondrial function such as decreases in the expression of PGC-1α, AMPK, and SIRT-1.

Taken together, these examples of studies from the zebrafish model highlight the potential of zebrafish in pioneering novel pathways and molecules for improving the mitochondrial function with the aim of developing therapies. In addition, while a lot of interventions such as supplementation with NAD as well as other dietary improvements have targeted larger rodent models like mice, none have been tested on zebrafish for assessing the improvement of aged muscle, despite the many advantages of the zebrafish model highlighted in this review and in many others. Therefore, focus should now be provided to the zebrafish model for understanding pathways related to mitochondrial function.

### 5.2. Validation of Novel Genes and Candidates Using the Zebrafish Animal Model

Genome-wide associations (GWAS) identifies genes and variants associated with or causal of disease, yet we need animal models to identify and decipher their functions in vivo. Regarding sarcopenia, there is a strong and substantial heritability, estimated to be 50%, in the transmission of defective genes through the genetic component [100]. Because there are no reliable markers to identify sarcopenia, GWAS studies are very much justified such that they can be used to identify single nucleotide polymorphisms (SNPs) related to sarcopenia. An additional advantage relating to sarcopenia and GWAS studies is that as a teleost model species, its genome has already been sequenced [3]. A key strength in the zebrafish genome is the existence of gene ohnologs or gene duplicates [3]. The zebrafish ohnologs have been found to be advantageous in the study of muscle diseases such as sarcopenia because they result in mild phenotypes, when deleted, as opposed to embryonic lethality observed in mammalian paralogs [3]. Although not related to sarcopenia, much recently, GWAS studies are now being employed in zebrafish models [101], and they have already enabled the identification of novel muscle pathways such as MPP7 that affect bone mineralization [102].

Moreover, with the advancements in gene-expression profiling, CRISPR gene editing, and the application of advanced computational models such as machine learning or artificial intelligence, among others, an increasing number of genes and proteins will likely be identified to have a potential role in the development and treatment of primary sarcopenia. The zebrafish, with its small genome size and ohnologs, emerges as an ideal candidate for utilizing these new tools in the identification and validation of novel genes and candidates associated with primary sarcopenia.

### 5.3. Complimentary with Other Animal Models

The short-lived multifaceted Turquoise Killifish has continued to rapidly emerge as a more practical experimental model for studying aging and diapause [103,104,105]. The use of this animal model will increase in the next decades as a complementary model to the current well-established vertebrate models like zebrafish and mice, as the Turquoise Killifish exhibits a short lifespan (4–6 months vs. 2–3 years) and a long diapause state [103,106]. It was demonstrated that the short-lived Turquoise Killifish upregulated H3K27 methylation complexes, leading to chromatin hoarding, and downregulated H3K27 acetylation complexes in the aging process [107]. Recently, a comparative study between the short-lived Turquoise Killifish and the longer-lived zebrafish demonstrated an initial central regulation of the NPY neuropeptide in aging models, with peripheral localization being present in both ageing models, elucidating new mechanisms that warrant further study in both fish species [108].

## 6. Conclusions

Recent advances in the zebrafish animal model have highlighted the potential role of this organism in elucidating the pathogenesis, diagnosis, monitoring, and treatment of sarcopenia. As the deterioration of muscle function is expected with aging, a better understanding of the impact of aging on muscles can help not only in managing sarcopenia but also in reducing the muscle decay that occurs with age. The gradual implementation of advanced molecular techniques in zebrafish and the recent validation of numerous models of sarcopenia will build the foundation for a better understanding and an improved management of sarcopenia in the upcoming decades.

## Figures and Tables

**Figure 1 ijms-24-17018-f001:**
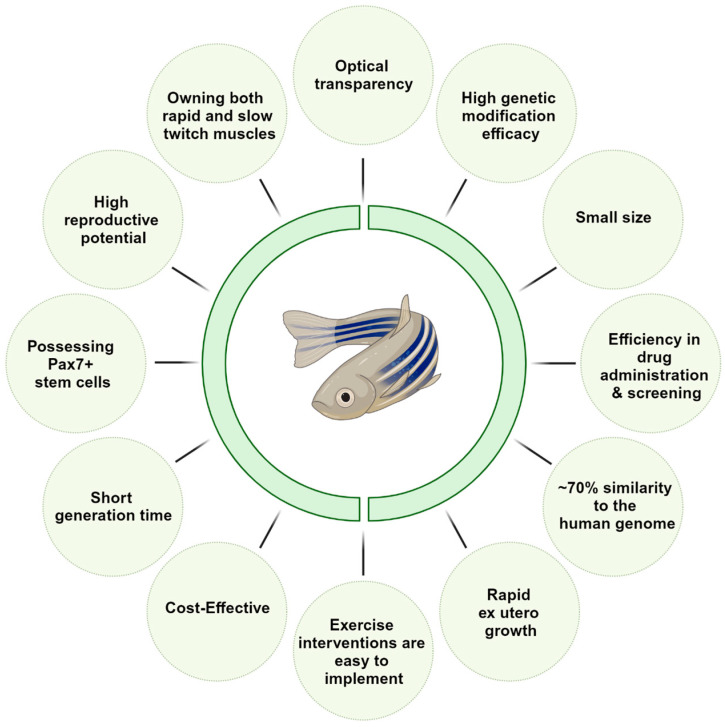
Summary of the numerous advantages of the Zebrafish model for unraveling the mechanistic pathways and potential therapies of primary sarcopenia.

**Figure 2 ijms-24-17018-f002:**
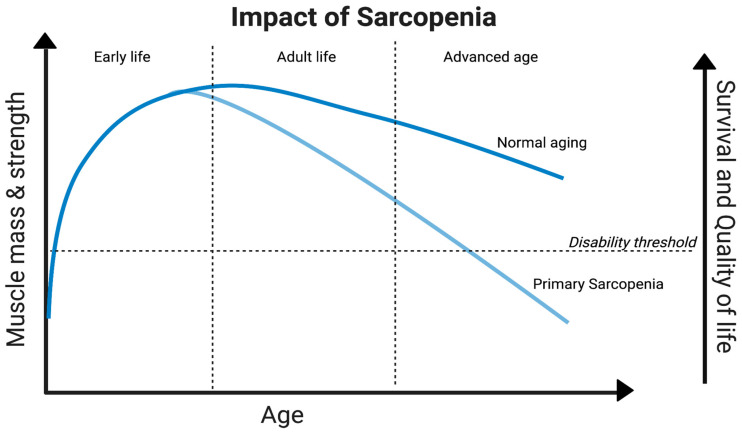
Deterioration of muscle aging and function throughout the lifespan of an individual. Zebrafish as a sarcopenia model would not only be beneficial to those with the disease but also to understand the mechanisms leading to muscle degradation in all individuals throughout their lifespan.

**Figure 3 ijms-24-17018-f003:**
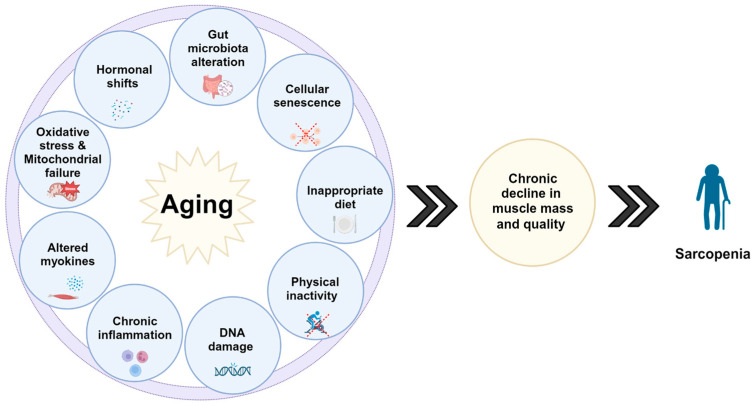
Multiple pathways lead to the deterioration of organ systems and ultimately lead to the worsening of muscle mass and function, impacting the functionality of an individual and leading to primary sarcopenia.

**Table 1 ijms-24-17018-t001:** Latest experimental studies in zebrafish as a sarcopenia model.

Title	Zebrafish Animals Used	Intervention and/or Exposure	Results	References
Electrical impedance myography detects age-related skeletal muscle atrophy in adult zebrafish	Wildtype casper (6 and 33 months) and *Tübingen* (4 and 24 months) zebrafish	Relationship between swimming efficacy, age, and EIM measures	EIM effectively correlates with age-related muscle atrophy in adult zebrafish.	[81]
Exercise intervention mitigates zebrafish age-related sarcopenia via alleviating mitochondrial dysfunction	AB strain male zebrafish (21 and 6 months)	Swimming efficacy on mitochondrial homeostasis and protein regulation in sarcopenia	Exercise reduced age-related muscle atrophy by improving muscle structure, decreasing protein breakdown, and restoring mitochondrial activity.	[95]
A high-fat diet induces muscle mitochondrial dysfunction and impairs swimming capacity in zebrafish: a new model of sarcopenic obesity	Adult male AB strain zebrafish (4 months)	High-fat diet (16 weeks) induced sarcopenic obesity models and exhibited effects on swimming capacity and muscle atrophy	High-fat diet for an extended period of time resulted in muscular atrophy, reduced swimming ability, increased body weight, higher muscle triglycerides, fatty liver characteristics, and the downregulation of mitochondrial and fatty acid metabolism genes.	[61]
Surface electrical impedance myography detects skeletal muscle atrophy in aged wildtype zebrafish and aged gpr27 knockout zebrafish	Tübingen-strain wild type (8, 12, and 36 months) and *gpr27* knockout zebrafish	Muscle atrophy detection in young, aged, and mutant zebrafish using surface EIM (sEIM)	sEIM effectively recognized muscle structural changes.	[94]
Aerobic exercise enhances mitochondrial homeostasis to counteract D-galactose-induced sarcopenia in zebrafish	Wild-type male AB strain zebrafish (7 months)	Correlation between D-galactose-induced sarcopenia models and aerobic activity	Aerobic exercise enhances muscle function and quality by regulating miR-128/IGF-1 pathway and improves mitochondrial homeostasis in aging muscle.	[96]
Chronic vitamin E deficiency dysregulates purine, phospholipid, and amino acid metabolism in aging zebrafish skeletal muscle	Zebrafish (55 dpf) supplemented for 12 or 18 months	Vitamin E-inadequate and -adequate diet for 12 or 18 months	The metabolic pathway alterations in skeletal muscle observed with aging and vitamin E deprivation exhibit some similarities but also demonstrate distinct modifications.	[59]

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
