# Peer review of "Zebrafish as an Emerging Model for Sarcopenia: Considerations, Current Insights, and Future Directions"

_ijms, 2023, doi:10.3390/ijms242317018_

Round 1

Reviewer 1 Report

Comments and Suggestions for Authors

The review paper Zebrafish as an Emerging Model for Sarcopenia: Considerations, Current Insights, and Future Directions aims to explore the unique advantages of zebrafish as a model for sarcopenia research, highlight recent breakthroughs, outline potential avenues for future investigations, and emphasize the distinctive contributions that zebrafish models offer. The paper presents adequate references and background research and the main contents are in line with the readers interest of IJMS. However, there are still some shortcomings that need to be further improved.

Comments:

Q1. The advantages and disadvantages of sarcopenia in other experimental animals and zebrafish still need to be further improved, since the assertion by certain researchers is unwavering that zebrafish, being non-mammalian organisms, exhibit significantly distinct mechanisms of action compared to those observed in humans.

Q2. The level of detail in many sub-headings of the article is insufficient, and it is recommended to enhance them further.

Q3. 1. Introduction is missing.

Q4. In sections 2. Zebrafish as an Animal Model of Sarcopenia and 3. Current challenges and opportunities, the contents in these two sections exhibits significant overlap, indicating the need for further refinement in terms of classification and distribution.

Author Response

The authors want to thank the reviewer´s comments, which greatly enhanced the quality of the manuscript. We made the following changes according to the reviewer:

Q1. The advantages and disadvantages of sarcopenia in other experimental animals and zebrafish still need to be further improved, since the assertion by certain researchers is unwavering that zebrafish, being non-mammalian organisms, exhibit significantly distinct mechanisms of action compared to those observed in humans.

A new paragraph was added to the manuscript in this section. We also think this paragraph helps address Q4 by separating the goals of each section of the manuscript. 

Q2. The level of detail in many sub-headings of the article is insufficient, and it is recommended to enhance them further.

Q3. “1. Introduction” is missing.

Thank you for pointing this out. The subheadings were edited and organized.

Q4. In sections “2. Zebrafish as an Animal Model of Sarcopenia” and “3. Current challenges and opportunities”, the contents in these two sections exhibit significant overlap, indicating the need for further refinement in terms of classification and distribution.

We agree with the reviewer. Adding the paragraph addressing Q1 further separated the objectives of each section and brought a more organized version of the text.

Reviewer 2 Report

Comments and Suggestions for Authors

1. In section 3, some advantages in using the zebrafish model for different aspects of sarcopenia research were not discussed in detail. For example, how zebrafish presents a distinct path to hope? How their potential utility in unraveling the complexity of sarcopenia and its underlying processes offers a viable path for cutting-edge treatments and therapeutic approaches?

2. In section 4, several latest advances in sarcopenia works using zebrafish, as listed in Table 1, are not discussed or only briefly discussed.

3. There are several reviews on fish models for human skeletal diseases. Besides the 6 articles published this year (as listed in Table 1), what is the update of this review in term of fish models for sarcopenia research?

4. In the perspectives, besides GWAS, other potential approaches that can be used for identification of novel genes should be discussed.

5. Lines 66-68, authors did not provide details on how the databases were systematically searched.

Author Response

The authors would like to thank the reviewer´s comments, which we think have greatly enhanced our manuscript. In response to the comments, we have made the following changes:

  1. In section 3, some advantages in using the zebrafish model for different aspects of sarcopenia research were not discussed in detail. For example, how zebrafish presents a distinct path to hope? How their potential utility in unraveling the complexity of sarcopenia and its underlying processes offers a viable path for cutting-edge treatments and therapeutic approaches?

Thank you for pointing this out. We added and developed further the section to better pinpoint the potential role of Zebrafish and the significant advantages available in this model.

  1. In section 4, several latest advances in sarcopenia works using zebrafish, as listed in Table 1, are not discussed or only briefly discussed.

We concur with the reviewer's comments and have addressed them by incorporating additional remarks in the manuscript, particularly emphasizing the potential roles, especially in the metabolism area, as highlighted by the majority of articles in the table. While we believe that the specific details of each article are more appropriately presented in the table, we are open to incorporating some details into the text if the reviewer deems it preferable in a subsequent revision.

  1. There are several reviews on fish models for human skeletal diseases. Besides the 6 articles published this year (as listed in Table 1), what is the update of this review in term of fish models for sarcopenia research?

We acknowledge the reviewer's observation regarding the availability of numerous reviews on zebrafish muscle research and various zebrafish models for muscle disorders. However, during the literature review, we found only one article specifically addressing zebrafish as a model for primary sarcopenia. Notably, this article predates the introduction of several studies listed in Table 1 and had a different scope, as practical zebrafish models for sarcopenia were not established at that time. Given the recent growth in opportunities, with studies uncovering different mechanisms and the emergence of novel high-output models like electrical impedance for assessing zebrafish muscle, we believe our review is timely and pertinent to the current state of the field, highlighting both existing knowledge and future opportunities in the coming decades.

  1. In the perspectives, besides GWAS, other potential approaches that can be used for identification of novel genes should be discussed.

We agree with the reviewer's comments and have incorporated a new paragraph into the manuscript, emphasizing the significance of novel technologies beyond GWAS and elucidating how zebrafish could play a pivotal role in leveraging these advancements

  1. Lines 66-68, authors did not provide details on how the databases were systematically searched.

Thank you for bringing this to our attention. We have expanded the paragraph to provide further clarification on our literature search methodology. Since our article is not a systematic review, we opted not to emphasize this section extensively. However, we did include information about the databases searched, a broad overview of the search strategy, and a general description of the inclusion criteria.

Reviewer 3 Report

Comments and Suggestions for Authors

The Manuscript: ‘Zebrafish as an Emerging Model for Sarcopenia: Considerations, Current Insights, and Future Directions’ by Santiago Callegari and colleagues explore the advantages of zebrafish as a model for sarcopenia research thereby highlighting recent breakthroughs in aging research through zebrafish model.

The study of sarcopenia, the age-related loss of muscle mass and strength, has gained significant attention in the field of aging research. While various animal models have been used to investigate the underlying mechanisms of sarcopenia, zebrafish have emerged as a promising and valuable model organism for studying this condition. Zebrafish, with its unique biological characteristics and genetic similarities to humans, provides a powerful platform for investigating the mechanisms of sarcopenia. Continued research using zebrafish models holds the key to unraveling the complexities of this age-related condition, ultimately leading to innovative therapies and improved quality of life for the elderly population. The submitted manuscript has contributed in this aspect.

After going through the manuscript, I have following comments for the authors:

1.     There are few recent studies analysing the molecular pathways associated with muscle wasting in zebrafish. A brief discussion of this aspect is recommended.

2.     Translating findings from zebrafish research to clinical applications holds immense potential for developing personalized interventions for sarcopenic patients. Please elaborate this understanding in the manuscript.

3.     Heading 1. Introduction is missing in the manuscript.

4.     Description/footnote of the figures should be beneath the figures and not above. As a thumb rule description of the figures is beneath and Tables is above.

5.     In table 1, the authors column (2nd column) is unclear. Are the works of the authors already listed in the reference section? If yes, I would suggest citing the references in standard form (As it is done in the rest of the manuscript; numerical citation).

Comments on the Quality of English Language

Moderate grammitical corrections and syntax errors need to be mended.

Author Response

We would like to thank the reviewer for the comments, which provided valuable insights and improved the manuscript. All the authors agree with the comments from the reviewer and incorporated all the suggested changes.

Round 2

Reviewer 2 Report

Comments and Suggestions for Authors

The authors have addressed all points raised in my previous review.